# Molecular and Clinical Aspects of Osteogenesis Imperfecta Type VI: A Case Series with Novel *SERPINF1* Gene Variants

**DOI:** 10.3390/ijms26136200

**Published:** 2025-06-27

**Authors:** Elena S. Merkuryeva, Tatyana S. Nagornova, Vladimir M. Kenis, Anna S. Deviataikina, Daria B. Akimova, Dmitry S. Buklaev, Ilya S. Dantsev, Aisluu O. Dulush, Ekaterina Y. Zakharova, Tatiana V. Markova

**Affiliations:** 1Research Centre for Medical Genetics, 115522 Moscow, Russia; elena.merkureva@gmail.com (E.S.M.);; 2The Turner Scientific Research Institute for Children’s Orthopedics, 196603 Saint Petersburg, Russia; 3Veltischev Research and Clinical Institute for Pediatrics, Pirogov Russian National Research Medical University, 125412 Moscow, Russia; 4Perinatal Center of the Republic of Tyva, 667000 Kyzyl, Russia

**Keywords:** osteogenesis imperfecta type VI, *SERPINF1* gene, bone fragility, fractures, founder effect, bone mineral density, genetic screening

## Abstract

Osteogenesis imperfecta type VI is a rare autosomal recessive disorder characterized by bone fragility and defective mineralization, caused by pathogenic variants in the *SERPINF1* gene. This study aimed to expand the understanding of OI type VI by analyzing clinical, radiological, and molecular findings in four patients from three unrelated families. Genotyping revealed two novel *SERPINF1* variants, c.185G>T (p.Gly62Val) and c.992_993insCA (p.Glu331Asnfs), in a compound heterozygous state in one patient, and a known pathogenic variant, c.261_265dup (p.Leu89Argfs26), in a homozygous form in three patients. Clinical manifestations included early-onset fractures, severe skeletal deformities, impaired mobility, and growth failure. Radiological assessments revealed multilevel and multiplanar bone deformities and metaphyseal widening. RNA analysis demonstrated that the c.992_993insCA variant results in a truncated PEDF protein without triggering nonsense-mediated decay. Population screening identified a carrier frequency of 0.0044 for the c.261_265dup variant, suggesting a founder effect in the Tuvinian population. These findings expand the mutational spectrum of the *SERPINF1* gene and provide new insights into the phenotypic variability of OI type VI. Our results highlight the importance of genetic screening in isolated populations and emphasize the need for further research to develop more effective therapeutic approaches for patients with limited response to bisphosphonate therapy.

## 1. Introduction

Osteogenesis imperfecta (OI) represents a clinically and genetically heterogeneous group of connective tissue disorders, the primary manifestation of which is increased bone fragility leading to multiple fractures and skeletal deformities [1]. To date, 23 genetic variants of OI, corresponding to types I–XXIII, have been described, and 21 causative genes have been identified, according to the OMIM database [2]. Among them, OI type VI (OMIM: 613982) occupies a special place, being a rare autosomal recessive disorder caused by pathogenic variants in the *SERPINF1* gene and characterized by a unique pathogenetic mechanism distinct from other OI types.

The *SERPINF1* gene, located on chromosome 17p13.3, encodes the glycoprotein pigment epithelium-derived factor (PEDF), which exhibits multifunctional properties, including the ability to modulate osteoclastogenesis [3,4]. The primary anti-osteoclastogenic mechanism of PEDF involves the stimulation of osteoprotegerin (OPG) expression and the inhibition of the RANKL signaling pathway, thereby contributing to the maintenance of the balance between bone formation and resorption [5]. Pathogenic variants in the *SERPINF1* gene disrupt the synthesis or function of PEDF, leading to increased osteoclast activity and, consequently, rapid bone loss. Clinically, this manifests as a severe course of OI, with multiple fractures occurring in infancy or early childhood, pronounced deformities of the long tubular bones, marked muscular hypotonia, and limited motor activity. Morphometric and radiological studies often reveal reduced bone mineral density, while histological examination identifies characteristic “fish-scale” lamellar bone and delayed osteoid maturation, features that morphologically distinguish type VI OI from other forms of the disease [6].

Since the first identification of pathogenic variants in the *SERPINF1* gene as the cause of OI type VI in 2011, fewer than 50 patients with this form of the disease have been reported in the scientific literature [3]. According to the LOVD database, 56 variants in the *SERPINF1* gene have been registered, of which 51 are classified as pathogenic and 5 as likely pathogenic [7].

Given the rarity of this OI type, the gradual accumulation of clinical data on patients carrying newly identified and previously described *SERPINF1* variants is crucial for deepening our understanding of the disease’s clinical course and pathogenesis. Such knowledge is essential for optimizing patient management and improving the quality of genetic diagnostics.

The aim of the present study is to analyze the molecular genetic and clinical–radiological features in four patients from three unrelated families with OI type VI: one patient carrying novel compound heterozygous variants in the *SERPINF1* gene, and three patients of Tuvan ethnicity harboring a previously described homozygous variant.

## 2. Results

### 2.1. Patient Report

Patient 1. A 23-year-old male proband. He was examined due to complaints of pathological fractures, skeletal deformities, muscle pain, short stature, and inability to walk independently. He was born from the first pregnancy, which was complicated by gestational issues. Delivery occurred at 42 weeks of gestation in breech presentation. His birth weight was 3450 g, and his length was 51 cm. The parents were healthy and of Russian ethnicity. Early psychomotor development was within normal limits.

The first fracture (femur) occurred at the age of 13 months following minimal trauma. Subsequently, up to 50 fractures were recorded by the age of 6 years, and approximately 200 by the age of 10 years. Deformities of the long tubular bones were noted starting at 3 years of age. Independent ambulation persisted until the age of 5 years, after which the patient required assistance. By 10 years, he was wheelchair-bound. Treatment with pamidronic acid was initiated at the age of 6, and denosumab therapy was introduced at 18 years. Dual-energy X-ray absorptiometry (DEXA) of the lumbar spine revealed reduced bone mineral density (BMD), with Z-scores (standard deviation scores used in densitometry) ranging from −1.5 to −3.1. Multiple orthopedic interventions were performed, including osteosynthesis, corrective osteotomies, and multilevel spinal reconstruction. Growth retardation became pronounced starting at 8 years of age.

At the time of examination, the height deficit was −5.52 standard deviations (SD). Physical examination revealed a broad, funnel-shaped chest, thoracolumbar scoliosis, deformities of the upper and lower limbs, saber-like deformity of the left forearm (due to previous fractures), and marked muscular hypotonia (Figure 1A,B).

Radiographic examination of the lower limbs in the anteroposterior view at the age of 11 years revealed a multiplanar (varus-antecurvatum) deformity of both femora, secondary to multiple pathological fractures. A Looser zone (stress fracture) was observed in the midshaft of the left femur. In Patient 1, surgical treatment involved intramedullary fixation of the right femur, while the left femur was immobilized using a plaster cast. Additional findings included deep acetabulae, deformation of the pelvic ring with vertically elongated obturator foramina, and multiple dense transverse metaphyseal bands (“zebra lines”) related to previous bisphosphonate therapy (Figure 2A).

At the age of 17 years, radiographs (Figure 2B) demonstrated restored axial alignment of both femora following surgical correction with telescopic intramedullary nails. Radiographic evaluation of the right humerus (Figure 2C) showed a pathological fracture treated with intramedullary nailing. Delayed bone union was noted, with the formation of a stiff pseudoarthrosis. Examination of the right forearm (Figure 2D) revealed a multiplanar (S-shaped) deformity of the radius and ulna, as well as a stiff pseudoarthrosis of the ulnar midshaft. Multiple “zebra lines” in the metaphyseal regions were again evident, consistent with long-term bisphosphonate treatment.

Patient 2. A 3-year-old male proband. He was examined due to complaints of frequent pathological fractures and delayed physical development. The parents are healthy, deny consanguinity, and are of Tuvan ethnicity, residing in the Republic of Tyva (Southern Siberia, Russia). There are older siblings in the family, and the third pregnancy ended in early miscarriage. The proband was born from the fourth pregnancy, which was uneventful. Delivery was full-term at 39 weeks of gestation. Birth weight was 3422 g, length was 50 cm, and Apgar scores were 7/8. Early motor development was initially age-appropriate: he achieved head control at 2 months and independent sitting at 6 months. Subsequently, a delay in motor development became evident due to muscular hypotonia: the child was unable to roll over, crawl, or walk.

The first pathological fracture (right femur) occurred at the age of 6 months. At 9 months, repeated fractures of the right femur and right humerus were registered. By the age of 1 year and 9 months, more than 15 fractures had been documented, occurring without apparent trauma. Zoledronic acid therapy was initiated at 1 year and 8 months. Bilateral intramedullary fixation of the femora was performed.

At the time of examination, the patient exhibited a keel-shaped chest deformity, combined deformities of the upper and lower limbs, growth retardation (body length −2.73 SD), skeletal deformities, and muscular dystonia (Figure 1C,D). The child was unable to sit, stand, or walk; he moved using a wheelchair and required support for sitting and standing. A delay in speech development was also noted. Surgical implantation of expandable intramedullary rods for stabilization of the long bones was planned.

Patient 3. A 16-year-old female proband. She was examined due to complaints of frequent pathological fractures, severe skeletal deformities, and inability to walk independently. She was born from the second pregnancy at 39 weeks of gestation via full-term delivery. Birth weight was 3180 g, length was 50 cm, and Apgar scores were 8/9. The parents are healthy, of Tuvan ethnicity, and deny consanguinity. There are four children in the family. The youngest brother (12 years old) has the same condition (Patient 4), while the two other siblings are healthy.

The first fracture (left humerus) occurred at the age of 1 year and 6 months. Subsequent fractures were recorded (a total of 12), including a clavicle fracture at 5 years, a compression fracture of Th12 at 7 years, and bilateral femoral fractures at 7 years requiring surgical treatment. Until the age of 7 years, she was able to walk independently and attended school. Since the age of 9 years, she has been wheelchair-bound. Corrective osteotomies with fixation using telescopic intramedullary rods have been performed.

At the last follow-up at 16 years of age, short stature (−2.96 SD) was noted. Physical examination revealed kyphotic deformity of the thoracic cage and varus deformities of the humeri (Figure 1E). Range of motion in the shoulder, elbow, and wrist joints was preserved, with the hands in a neutral position. The lower limbs exhibited a length discrepancy (right shorter than left by 4 cm), contractures of the hip joints, and limited flexion (up to 120°) and extension (up to 165°) in the left knee. The mechanical axis of the right lower limb was preserved, and the feet were in a neutral position. The patient demonstrated extreme caution during movements due to fear of new fractures. She was unable to walk independently and required assistance (following surgery on the left lower leg).

In proband 3, lateral radiography (Figure 2E) of the thoracolumbar spine revealed mild pseudo-butterfly deformities of the lower thoracic and upper lumbar vertebral bodies, accompanied by thoracolumbar kyphosis. Anteroposterior views (Figure 2F) demonstrated a bell-shaped thorax with thin, vertically oriented ribs and signs of multiple healed rib fractures. Mild scoliosis was also noted. The humeri (Figure 2G,H) showed varus deformities, contributing to the functional limitations of the upper extremities. In proband 3, postoperative radiographs of the lower limbs following correction with telescopic intramedullary rods demonstrated improved axial alignment (Figure 2I). However, residual skeletal abnormalities persisted, including coxa vara on the left with delayed union, protrusio acetabuli, and a deformed pelvic ring with vertically elongated obturator foramina.

Patient 4. A 12-year-old boy, the younger brother of Patient 3. He was examined due to delayed motor development and multiple pathological fractures. He was born from the third pregnancy by full-term delivery at 40–41 weeks of gestation. Birth weight was 3180 g, length was 50 cm. A delay in motor milestones was noted: head control was achieved at 4 months. The first fracture (right femur) occurred at the age of 9 months. He never acquired the ability to sit or walk independently. A total of 14 fractures were recorded. Since the age of 6 years, he has been receiving intermittent bisphosphonate therapy.

At the time of examination, the patient exhibited a severe growth deficit (−7.71 SD). Kyphoscoliotic spinal deformity, a keel-shaped chest (characterized by anterior protrusion and posterior flattening), and clavicular deformities were observed (Figure 1F). The upper limbs demonstrated multiplanar deformities with marked shortening. Active movements in the shoulder and elbow joints were restricted, while passive movements were painful. Although the hands were fixed in a neutral position, grasping ability was preserved. The lower limbs were externally rotated, with antecurvation and varus deformities of the femora, as well as antecurvation deformities of the tibiae. Attempts to reposition the limbs to a neutral alignment elicited moderate pain. Flexion contractures of the knee joints and plano-valgus deformity of the feet were also present. The patient was unable to sit or walk and remained predominantly in a lying position.

Three-dimensional reconstruction of the whole body of Patient 4 (Figure 2J) revealed multiple multilevel, multiplanar deformities of the long tubular bones of both the upper and lower extremities, resulting from fractures and pathological bone remodeling, with deformity apices predominantly located in the diaphyseal regions. The axes of the knee and elbow joints, as well as the juxta-articular regions (epiphyses and metaphyses), were relatively preserved. A pronounced deformation and narrowing of the pelvic ring were observed due to acetabular protrusions, along with vertically oriented obturator foramina and pseudo-butterfly vertebrae. Deformities of the spine (kyphoscoliosis) and thoracic cage were noted, along with multiple rib fractures at various stages of healing and callus remodeling. No significant asymmetry or deformity of the craniofacial and neurocranial bones was observed. No Wormian bones were detected (Figure 2J).

The main clinical manifestations observed in the examined patients included multiple fractures, severe deformities of the long tubular bones, short stature, and reduced bone mineral density according to densitometry data. Information on the patients, including demographic, clinical, and radiological characteristics, as well as the age at initiation of bisphosphonate therapy, is summarized in Table 1.

The age at first fracture among the examined patients ranged from 6 to 18 months. Femoral fractures were the most frequently observed. All patients demonstrated decreased vertebral body height on spinal radiographs, indicative of vertebral fractures of varying severity. The total number of fractures and the annual fracture rate were highest in Patients 1 and 2. Additionally, Patient 1 required surgery to correct severe scoliosis (grade IV), which was performed at the age of 17 years. All patients exhibited very short stature (ranging from −2.73 to −7.71 SD). Bone mineral density, measured in the lumbar spine, was the lowest in Patient 4. No clinical signs of hearing loss, visual impairment, or dentinogenesis imperfecta were observed in the examined patients with OI type VI. None of the patients were able to walk independently, and Patients 2 and 4 were unable to sit without support. Although the small sample size precludes definitive conclusions, the data suggest potential genotype–phenotype trends warranting further investigation.

### 2.2. Molecular Genetic Analysis

The diagnosis of OI type VI was confirmed in the examined patients by sequencing a targeted gene panel including genes responsible for OI (see Appendix A). In the *SERPINF1* gene (NM_002615.7), two novel variants and one previously described variant were identified in four patients from three unrelated families with OI type VI. The newly identified variants were detected in Patient 1 in a compound heterozygous state: a likely pathogenic missense variant in exon 3, the c.185G>T variant, resulting in a missense substitution (p.Gly62Val), was classified as likely pathogenic according to ACMG criteria (PP3, PM5, PM2, PP4), and in exon 7, a pathogenic insertion c.992_993insCA, leading to a frameshift (p.Glu331Asnfs), was identified (PVS1, PM2, PP4). Sanger sequencing confirmed the presence of both variants in the proband and revealed that the missense variant (p.Gly62Val) was inherited from the mother, while the frameshift variant (p.Glu331Asnfs) was inherited from the father (Figure 3).

In three patients (Patients 2, 3, and 4) from two unrelated Tuvan families, a homozygous duplication c.261_265dup was identified, resulting in a frameshift, the substitution of leucine at position 89 with arginine, and a subsequent premature stop codon after 26 amino acids (p.Leu89Argfs*26). Heterozygous carriage of this variant was confirmed in the healthy parents and siblings.

### 2.3. RNA Analysis of the Variant c.992_993insCA

To assess the impact of the c.992_993insCA variant on protein function, RT-PCR analysis of total RNA extracted from the patient’s skin fibroblasts was performed, followed by deep sequencing of the PCR products. The analysis revealed no alterations in the mRNA structure of the *SERPINF1* gene or evidence of allelic imbalance. It was demonstrated that, at the protein level, the c.992_993insCA insertion leads to a truncation of the protein by 86 amino acids (p.Glu331AsnfsTer3), which is associated with a severe disease phenotype in the patient.

### 2.4. Frequency Assessment of Type VI Osteogenesis Imperfecta in the Republic of Tuva

The duplication c.261_265dup (p.Leu89Argfs*26) was previously reported by Li L. et al. (2019) in two patients from unrelated families in China: in one patient in the homozygous state, and in the other in a compound heterozygous state with a deletion c.879del (p.Thr294Profs) [8]. In 2023, the same mutation was identified in the homozygous state in a patient from the Republic of Tyva [9]. Based on these findings, it has been suggested that the homozygous c.261_265dup mutation in the *SERPINF1* gene may be relatively common in the Tuvan population.

As a result of the analysis of 801 newborn dried blood spot (DBS) samples, seven heterozygous carriers of the c.261_265dup (p.Leu89Argfs*26) mutation in the *SERPINF1* gene were identified.

The estimated allele frequency was 0.0044 (95% confidence interval (CI): 0.0018–0.0090), corresponding to approximately 1 carrier per 114 individuals (95% CI: 1:284–1:56). The estimated disease frequency (q^2^) was 0.000019, or approximately 1 in 52,375 individuals (95% CI: 1:323,370–1:12,395).

## 3. Discussion

OI type VI was first described in 2002 by Glorieux and colleagues in eight patients with severe clinical manifestations, including frequent fractures, deformities of the long tubular bones, and low bone mineral density. This OI type was distinguished based on the histological features of bone biopsy specimens, which revealed a large amount of unmineralized osteoid tissue, foci of osteomalacia, and disorganization of the bone matrix, where the lamellar structure acquired a characteristic “fish scale” appearance [6,10]. In 2011, Becker and colleagues identified pathogenic variants in the *SERPINF1* gene associated with severe forms of OI type VI and confirmed the autosomal recessive mode of inheritance of this disorder [4]. Subsequent studies in various populations demonstrated significant variability in clinical manifestations despite the overall severity of the disease: in some patients, fractures occurred early in life, while in others they developed later, the degree of skeletal deformities also varied considerably, even among members of the same family carrying the same mutation [3,11,12,13,14]. In particular, cases of intrafamilial phenotypic variability have been reported, highlighting the complexity of phenotypic expression in the presence of identical genetic defects [14].

In the presented clinical cases, no physical abnormalities were observed at birth in any of the four patients. The earliest age at first fracture was recorded in Patient 2 at 6 months. In Patient 3, the form of OI initially appeared mild, with the first fracture occurring at 18 months. She was able to attend school and had no physical activity limitations until 9 years of age. However, as she grew older, the frequency of fractures increased, and deformities of the lower limbs progressively worsened.

At the same time, her full sibling (Patient 4) exhibited severe growth retardation (−7.71 SD at 12 years of age) and was never able to sit or walk independently due to pronounced muscular hypotonia, similarly to Patient 2. Importantly, despite conservative and surgical treatment, all examined patients demonstrated a progressively severe disease course, leading to pronounced deformities of the long tubular bones, thoracic cage, and spine, as previously described in other studies [4,6,15]. Two probands lost the ability to walk independently, and two others were bedridden and unable to sit.

Consistent with previous reports, patients with OI type VI exhibited white or slightly bluish sclerae, with no involvement of the dentition or hearing organs, a finding confirmed in all patients examined in this study [6].

The radiological features observed in our patients were generally consistent with those previously described for OI type VI [3,6,14,15]. Multiple multilevel and multiplanar deformities of the long tubular bones of both the upper and lower extremities were noted, resulting from pathological fractures. Characteristic findings included reduced bone mineral density, disruption of bone architecture, and cortical thinning. In Patient 2, metaphyseal widening with relative narrowing of the diaphyses was identified, likely associated with prolonged bisphosphonate therapy. Such changes have previously been referred to as “onion skin-like metaphyses”. These metaphyseal deformities reflect impaired normal bone remodeling and may be exacerbated by pharmacological interventions, particularly during long-term use of antiresorptive agents.

The protein product of the *SERPINF1* gene plays a key role in maintaining bone homeostasis. PEDF acts as an inhibitor of osteoclastogenesis by regulating osteoprotegerin (OPG) expression and suppressing RANKL-induced differentiation and proliferation of osteoclast precursors [3,5].

Pathogenic homozygous or compound heterozygous variants in *SERPINF1* lead to PEDF deficiency, resulting in a disruption of the RANKL/OPG system balance. This promotes excessive osteoclast activity and progressive bone loss—a key pathogenic mechanism in OI type VI.

Due to these mineral metabolism characteristics, bisphosphonate therapy has limited efficacy in such patients. It is assumed that the excessive amount of unmineralized osteoid tissue hinders bisphosphonate binding to the mineralized bone surface, thereby reducing their effectiveness in suppressing osteoclast activity. In contrast, agents such as denosumab act through a different mechanism, inhibiting osteoclast activation prior to their adhesion to the bone matrix, which may render them more effective in OI type VI. Previous studies by Semler et al., based on a small patient cohort, suggested that denosumab may be more effective in reducing bone resorption in patients with OI type VI compared to bisphosphonates [16,17]. In our cohort, only Patient 1 received denosumab therapy after the age of 18 and reported a significant reduction in pain under treatment. Currently, studies on the efficacy of denosumab in OI type VI are ongoing and require further accumulation of clinical data.

In the present study, novel variants in the *SERPINF1* gene: c.185G>T (p.Gly62Val) and c.992_993insCA (p.Glu331Asnfs) were identified in a compound heterozygous state in Patient 1, and a previously described variant, c.261_265dup (p.Leu89Argfs*26), was detected in a homozygous state in Patients 2, 3, and 4.

The missense variant c.185G>T (p.Gly62Val) in exon 3 of the *SERPINF1* gene results in an alteration of the amino acid sequence within the N-terminal domain of the PEDF protein, in a region that may influence secretory processes and structural stability. Although this position does not directly affect known anti-angiogenic or receptor-binding sites, changes in this region may indirectly impact the protein’s functions.

The insertion of two nucleotides (c.992_993insCA) in exon 7 of the *SERPINF1* gene results in a frameshift, leading to the emergence of a premature stop codon and truncation of the protein by 86 amino acids at the C-terminus (p.Glu331AsnfsTer3), corresponding to approximately 15–20% of the total protein length. Loss of the critical C-terminal region of PEDF disrupts the formation and function of an essential functional domain. It is known that the C-terminal region of PEDF is involved in binding to extracellular matrix proteins and is responsible for several biological functions, including collagen interaction and angiogenesis regulation [18,19]. Our RT-PCR analysis demonstrated that the mRNA carrying this insertion is stable, does not undergo apparent nonsense-mediated decay (NMD) or degradation, and no allelic imbalance was detected. Thus, the results show that the frameshift variant c.992_993insCA (p.Glu331AsnfsTer3) does not cause mRNA degradation but leads to the production of a truncated protein likely lacking a critical functional domain. In a compound heterozygous state with the missense substitution p.Gly62Val, this variant underlies the severe form of OI type VI observed in Patient 1.

The pathogenic duplication in exon 3 of the *SERPINF1* gene, c.261_265dup, p.(Leu89Argfs*26), in the homozygous state, leads to a complete or substantial loss of PEDF protein function due to a frameshift and premature termination of translation. This, in turn, results in severe impairment of angiogenesis, bone remodeling, and bone matrix organization, leading to pronounced bone fragility and a severe disease course in Patients 2, 3, and 4, consistent with previous reports [7,9].

To date, the prevalence of osteogenesis imperfecta (OI) type VI is known to vary significantly across different populations. In a large cohort study conducted in the Russian Federation at the Medical Genetic Research Center, OI type VI accounted for 10.3% of all rare forms of the disease with autosomal recessive inheritance [20]. In a Chinese cohort, this subtype ranked second among autosomal recessive forms of OI, reaching 22%, while in India it accounted for 12.5% among all OI subtypes [7,14]. Notably, in the Indian population, the pathogenic c.829_831del (p.Phe277del) variant in the *SERPINF1* gene is recurrent and exhibits a “founder effect”.

Our findings support previously reported features of the Tuvan gene pool, which is characterized by a “founder effect” [21]. A high carrier frequency of the pathogenic c.261_265dup variant in the *SERPINF1* gene (0.0044 or 1:114) was established, and the estimated prevalence of OI type VI in the Tuvan population was calculated to be approximately 0.000019 (1:52,375), highlighting the importance of targeted molecular screening in isolated ethnic groups. These data complement previous reports on the high prevalence X of other recessive pathogenic variants, such as c.516G>C, c.-23+1G>A, and c.235delC in the GJB2 gene, as well as c.125_126insTGGCG (p.Trp42CysfsTer12) in the NPR2 gene [21,22]. Such a concentration of pathogenic variants is likely attributable to the long-term isolation of the population, its relatively small size, and the preservation of common ancestral lineages across generations. These features underscore the need for further research aimed at identifying region-specific mutations and elucidating the mechanisms underlying the increased frequency of rare genetic disorders in the Tuvan population.

## 4. Materials and Methods

### 4.1. Clinical Evaluation

The study involved a thorough examination of four patients from three unrelated families, ranging in age from 3 to 23 years, who presented with clinical and radiological signs of OI. Patient #1 was of Russian descent, while patients #2, #3, and #4 were of Tuvan origin. Patients #3 and #4 were siblings. To establish a definitive diagnosis, a range of diagnostic procedures was employed, including genealogical analysis, clinical evaluation, and neurological assessment using a standardized method with a focus on the psycho-emotional sphere.

### 4.2. Radiographic Analysis

Radiographic evaluation included imaging of the spine, hip joints, and long bones of the extremities to identify skeletal abnormalities such as deformities, fractures, and bone density changes.

### 4.3. Genetic Sequencing

The methodological workflow, including sequencing and variant analysis steps, is illustrated in Figure 4.

To genotype the probands, targeted panel sequencing was conducted, encompassing 166 genes associated with hereditary skeletal disorders.

Genomic DNA was isolated from dried blood spots (DBS) using the DNeasy kit («QiaGen», Hilden, Germany) according to the manufacturer’s standard protocol. DNA concentration was measured using a Qubit 2.0 instrument with Qubit BR and Qubit HS reagents (Thermo Fisher Scientific, Waltham, MA, USA) following the manufacturer’s protocol. Sample preparation was performed using a multiplex polymerase chain reaction (PCR) method targeting specific DNA regions. Next-generation sequencing (NGS) was performed using an Ion Torrent S5 sequencer, achieving an average coverage of at least 80x, with targeted regions covered at a minimum of 20x in ≥90–94% of cases. Coverage uniformity was assessed to ensure data quality. Initial sequencing data analysis was conducted using the standard automated algorithm provided by Ion Torrent. RT-PCR assays were performed on the QuantStudio 5 PCR System (Applied Biosystems, Foster City, CA, USA). A fragment in exon 3 of the *SERPINF1* gene was amplified in both the research and control samples (5 unrelated women and 5 men without any bone diseases, selected based on the absence of skeletal disorders). Calibration curves were generated to quantify unknown samples. In patients with identified variants, a search for the second variant was performed using automated Sanger sequencing on the ABI Prism 3500xl device (Applied Biosystems, Waltham, USA) according to the manufacturer’s protocol. Primer sequences were designed based on the reference sequence of the *SERPINF1* gene target regions (NM_002615.7).

Identified variants were annotated using the nomenclature provided on the website http://varnomen.hgvs.org/recommendations/DNA (accessed on 20 January 2025) (version 2.15.11). Population frequencies of the identified variants were assessed using data from the 1000 Genomes Project, ESP6500, and the Genome Aggregation Database (gnomAD) v2.1.1 (see Appendix A) [23,24,25]. The clinical significance of the variants was evaluated using the OMIM database and the HGMD^®^ Professional pathogenic variants database (version 2021.3) [26]. Pathogenicity and causality of the genetic variants were assessed in accordance with international guidelines for the interpretation of data obtained through massive parallel sequencing.

Data evaluation was performed using the ABI Prism sequence detection system.

### 4.4. Population Study

To estimate the frequency of OI in the Republic of Tuva, 801 DBS samples from newborns collected at the Perinatal Center of the Republic of Tyva were analyzed using allele-specific real-time PCR. The analysis was carried out using the TaqMan^®^ Allelic Discrimination Assay [Applied Biosystems. TaqMan^®^ Allelic Discrimination Assay: User Guide; Thermo Fisher Scientific: Waltham, MA, USA, 2013. Available online: https://tools.thermofisher.com/content/sfs/manuals/cms_041280.pdf (accessed on 22 May 2025)].

Primers were designed to amplify a fragment of the *SERPINF1* gene flanking the c.261_265dup variant. Each reaction was performed in duplicate using two sets of primers:
SERPINF1_261_265dup_RT_1_F: AACGTGCTCCTGTCTCCTCT (0.04 µM)SERPINF1_261_265dup_RT_1_R: GAGCACTCACCCAGCGAG (0.04 µM)SERPINF1_261_265dup_RT_2_F: CTGTCTCCTCTCAGTGTGGC (0.04 µM)SERPINF1_261_265dup_RT_2_R: GCTTCCTGCATCTGAGCACT (0.04 µM)

Two TaqMan probes were used to anneal specifically to the wild-type and mutant alleles. PCR was performed on a QuantStudio 5 system using SmarTaq polymerase (DIALAT, Russia) with an optimized protocol. Raw data were analyzed using Design and Analysis Software v2.6 (QuantStudio family) and visualized as an allelic discrimination plot.

The prevalence of OI type VI was calculated using the Hardy–Weinberg equilibrium Equation (1):*p*^2^ + 2*pq* + *q*^2^ = 1(1)
where 2*pq* is the expected frequency of heterozygotes, and *q* is the allele frequency of the variant under investigation.

The observed heterozygous carrier frequency of the pathogenic SERPINF1 variant was calculated using Equation (2):2*pq* = *m*/2*N*(2)
where 2*pq* is the proportion of heterozygotes, *m* is the number of identified heterozygotes, and *N* is the total number of individuals screened.

The 95% confidence intervals (CI) for the mutant allele and disease frequencies were calculated using the Clopper–Pearson exact method, implemented via the BinomCI function in the DescTools package (version 0.99.59) in R [27].

### 4.5. RNA Analysis of the Variant c.992_993insCA

The mRNA structure was analyzed using RNA extracted from fibroblasts of the proband. The cell cultures were acquired and stored in the Moscow Branch of the Biobank, which is part of the “All-Russian Collection of Biological Samples of Hereditary Diseases.” RNA extraction was performed using the Extract RNA reagent (Evrogen, Russia).

Reverse transcription was carried out using the Reverse Transcription System (Dialat, Russia) following the manufacturer’s protocol. The reaction was prepared in a final volume of 25 μL containing 13 μL of total RNA, 4 μL of 5X MasterMix, and 1 μL of Oligo(dT)18 primer (0.5 μg/μL). The mixture underwent initial incubation at 25 °C for 10 min for primer annealing, followed by cDNA synthesis at 45 °C for 30 min. The reaction was terminated by enzyme inactivation at 85 °C for 5 min. The synthesized cDNA was stored at −20 °C until further analysis. The quality of the synthesized cDNA was verified through qPCR of housekeeping genes.

To evaluate the impact of the variant on mRNA structure, the target region of the SERPINF1 gene was amplified using specific primers: 5′-CTGGTGCTACTCCTCTGCAT-3′ and 5′-TCCTCGTTCCACTCAAAGCC-3′. The resulting product was then subjected to next-generation sequencing (NGS).

#### Targeted Next-Generation Sequencing of RT-PCR Product

NGS libraries were prepared using the “SG GM” Kit (Raissol) and sequenced on the FASTASeq platform in paired-end mode (2 × 150 bp). The targeted locus achieved a coverage exceeding 82,000× in the proband’s sample and over 120,000× and 87,000× in the control samples. The raw sequencing data were analyzed using a custom bioinformatics pipeline, which included quality control with FastQC v0.12.1, alignment to the hg38 human genome assembly using STAR 2.7.11b, and sorting of aligned reads. Splice junctions were visualized using Sashimi plots in the Integrative Genomics Viewer (IGV).

Informed consent was obtained from the parents or legal representatives of the patients for clinical examination and molecular genetic analysis. The study was approved by the local ethics committee (protocol code 2021-3, 12 March 2021), and permission was granted for the anonymous publication of the study results.

## 5. Conclusions

The obtained data confirm the crucial role of molecular genetic analysis in the diagnosis of OI type VI associated with variants in the *SERPINF1* gene. The newly identified variants expanded the spectrum of molecular defects and allowed for a detailed characterization of the clinical manifestations. These findings contribute to improving diagnostic strategies and genetic counseling, and also highlight the need for the development of novel therapeutic approaches, given the limited effectiveness of bisphosphonate therapy.

## Figures and Tables

**Figure 1 ijms-26-06200-f001:**
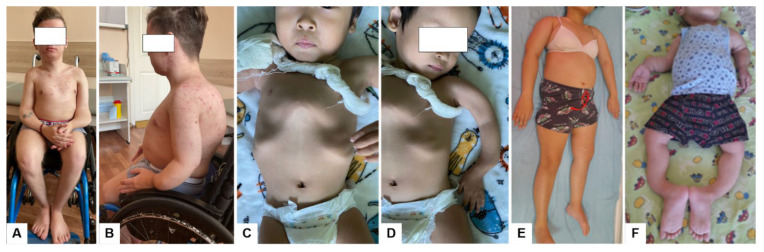
External clinical features of patients with osteogenesis imperfecta type VI: (**A**,**B**) Patient 1; (**C**,**D**) Patient 2; (**E**) Patient 3; (**F**) Patient 4.

**Figure 2 ijms-26-06200-f002:**
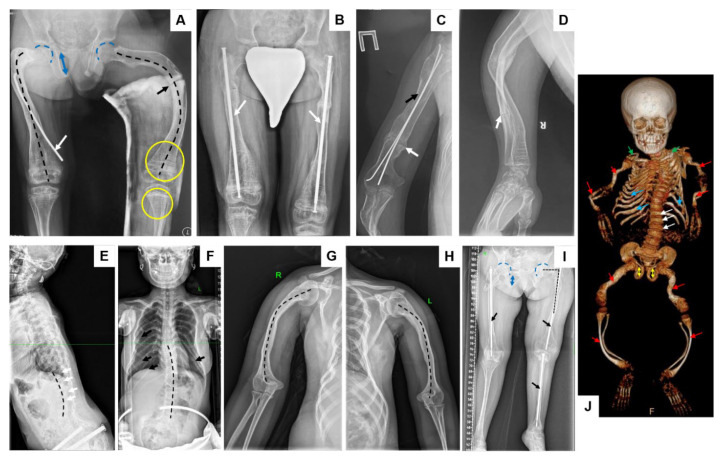
Radiographic findings in patients with osteogenesis imperfecta type VI:. (**A**) Anteroposterior radiograph of lower limbs of proband 1: multiplanar (varus-antecurvatum) deformity of both femora secondary to pathological fractures (black broken lines), treated surgically with intramedullary nail on the right side (white arrow) and plaster on the left side; Looser zone (stress fracture) of the midshaft of the left femur (black arrow); deep acetabulae (blue broken lines), deformed pelvic ring (vertically elongated obturator foramina (blue arrow), multiple “zebra-lines” in the metaphyseal regions after bisphosphonates treatment (yellow circles). (**B**) Anteroposterior radiograph of lower limbs of proband 1 after surgical correction with telescopic intramedullary nails (white arrows): restored axial alignment of the femoral bones. (**C**) Radiograph of the right humerus of proband 1: pathological fracture, treated surgically with intramedullary nails (black arrow), delayed union with formation of stiff pseudoarthrosis (white arrow). (**D**) Radiograph of the right forearm of proband 1: multiplanar (S-shaped) deformity of radius and ulna, stiff pseudoarthrosis of ulnar midshaft (white arrow); multiple “zebra-lines” in the metaphyseal regions after bisphosphonates treatment. (**E**) Lateral radiograph of thoracolumbar spine of proband 3: mild pseudo-butterfly deformity of lower thoracic and upper lumbar vertebral bodies (white arrow), thoracolumbar kyphosis (black broken line). (**F**) Anteroposterior radiograph of chest and thoracolumbar spine of proband 3: mild scoliosis (black broken line), bell-shaped thorax, thin, vertically oriented ribs, multiple healed rib fractures (black arrows). (**G**,**H**) Anteroposterior radiograph of the humerus bones of proband 3: varus deformity (black broken lines). (**I**) Anteroposterior radiograph of lower limbs of proband 3 after surgical correction with telescopic intramedullary nails (black arrows): improved axial alignment of the bones, coxa vara on the left side with delayed union (black broken lines), protrusio acetabulae (blue broken lines), deformed pelvic ring with vertically elongated obturator foramina (blue arrow). (**J**) 3D reconstruction of the whole body CT of proband 4: multilevel, multiplanar deformities of the long tubular bones (red arrows), mild scoliosis, bell-shaped thorax, thin, vertically oriented ribs, multiple healed rib fractures, deformed pelvic ring with vertically elongated obturator foramina (yellow arrows), pseudo-butterfly deformity of lower thoracic and upper lumbar vertebral bodies (white arrows), S-shaped deformity of the clavicles (green arrows).

**Figure 3 ijms-26-06200-f003:**
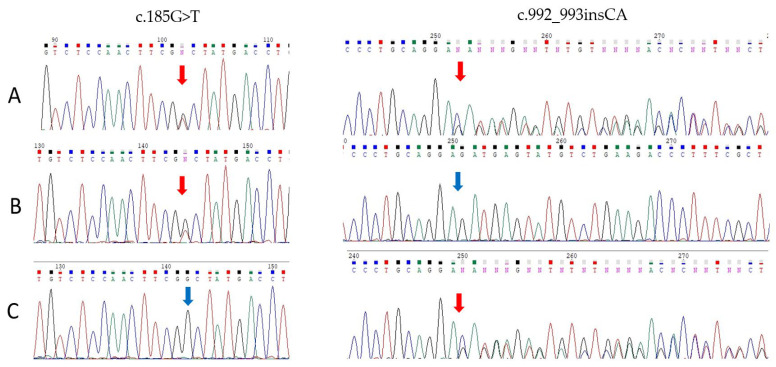
Sanger sequencing chromatograms of PCR products. Arrows indicate the positions of the identified mutations: missense (c.185G>T) and insertion (c.992_993insCA), in the proband (**A**), mother (**B**), and father (**C**).

**Figure 4 ijms-26-06200-f004:**
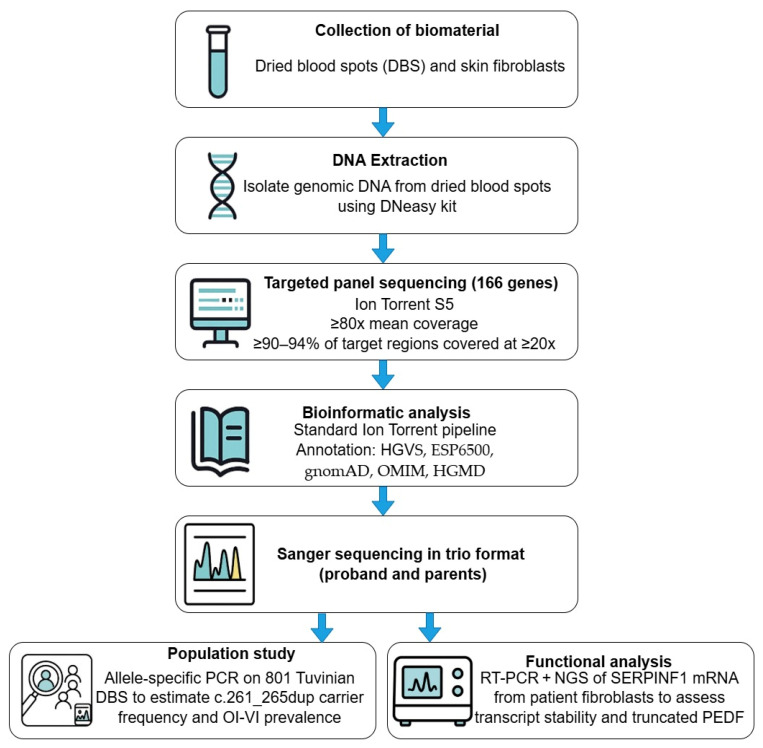
Study methodology: workflow of *SERPINF1* gene variant sequencing and analysis.

**Table 1 ijms-26-06200-t001:** Genotype–phenotype characteristics of four patients with osteogenesis imperfecta type VI caused by *SERPINF1* variants.

Characteristic	Patient 1	Patient 2	Patient 3 *	Patient 4 *
Mutation	c.185G>T/c.992_993insCA	c.261_265dup (homozygous)	c.261_265dup (homozygous)	c.261_265dup (homozygous)
Ethnicity	Russian	Tuvan	Tuvan	Tuvan
Consanguinity	No	No	No	No
Age (years)	23	3	16	12
Sex	M	M	F	M
Age at first fracture (months)	13	6	18	9
Fracture sites ^1^	1, 2, 3, 4, 5, 6	2, 3, 4, 5, 6	1, 2, 3, 4, 5, 6	1, 2, 3, 4, 5, 6
Fractures total (n)	200	15	12	14
Fractures/year	8.69	3.0	0.8	1.27
Scoliosis (grade)	IV	-	I	II
Severity of long bone deformities	Severe	Moderate	Moderate	Severe
Height (SD Score)	−5.52	−2.73	−2.96	−7.71
BMD (L1–L4) (g/cm^2^)	0.592	0.874	0.874	0.256
Z-score (L1–L4)	−3.1	−1.7	−1.7	−4.6
Blue sclera	No	No	No	No
Dentinogenesis imperfecta	No	No	No	No
Hearing loss	No	No	No	No
Age at initiation of bisphosphonates	6 years	1 year 8 months	8 years	6 years
Independent walking	No	No	No	No

***** Patients 3 and 4 are siblings from the same family. ^1^ Fracture sites: 1—rib, 2—humerus, 3—ulna and radius, 4—femur, 5—tibia and fibula, 6—vertebra. n — total number of fractures per patient; BMD—bone mineral density; SD—standard deviation.

## Data Availability

Data are contained within the article and Appendix A.

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
