# Peer review of "Molecular and Clinical Aspects of Osteogenesis Imperfecta Type VI: A Case Series with Novel SERPINF1 Gene Variants"

_ijms, 2025, doi:10.3390/ijms26136200_

Round 1
Reviewer 1 Report
Comments and Suggestions for Authors
Summary: The novelty of this manuscript is that highlights the crucial role of molecular genetic analysis in the diagnosis of OI type VI associated with variants in the SERPINF1 gene. As the authors stated, these finding highlights both the importance of genetic screening and the need for the development of novel therapeutic approaches because of the limited efficacy of the current therapies for OI type VI.
I have only minor comments:
- Page 7 “Sanger sequencing confirmed the presence of both variants in the proband and revealed that the missense variant (p.Gly62Val) was inherited from the mother, while the frameshift variant
(p.Glu331Asnfs) was inherited from the father”. No electropherograms were shown in the manuscript. Please, could you consider adding them.
- Page 10 “Primers were designed to amplify a fragment of the SERPINF1 gene flanking the c.261_265dup variant. Two TaqMan probes were designed to anneal to the wild-type and mutant alleles, respectively”. Please add primer sequences to analyze them.
- Page 6 “The diagnosis of OI type VI was confirmed in the examined patients by sequencing a targeted gene panel including genes responsible for OI”. Please describe in the Materials and Methods section the gene panel built for NGS analysis. This manuscript was original and very well written, and enriches our knowledge on a hot topic.
In my opinion, the format of the manuscript is in line with the journal’s criteria. The topic was well researched, and the references are appropriate and up-to-date.

Author Response
Response to the Reviewers We sincerely thank the reviewers for their detailed and insightful comments, which helped us to improve the clarity and quality of our manuscript. We have addressed all the suggestions and concerns raised, and corresponding changes have been made in the revised version of the manuscript. Below, we provide a detailed point-by-point response to each reviewer’s comments.
Minor сomments:
Page 7 “Sanger sequencing confirmed the presence of both variants in the proband and revealed that the missense variant (p.Gly62Val) was inherited from the mother, while the frameshift variant (p.Glu331Asnfs) was inherited from the father”. No electropherograms were shown in the manuscript. Please, could you consider adding them.
Response:
We thank the reviewer for this helpful comment. In response, we have added a new figure to the manuscript, presenting Sanger sequencing chromatograms that confirm the presence of two variants in the SERPINF1 gene:
· c.185G>T (p.Gly62Val) — in the proband and his mother (panels A and B),
· c.992_993insCA (p.Glu331Asnfs) — in the proband and his father (panels A and C).
Red arrows indicate the variant positions, while blue arrows highlight the parental inheritance. These data confirm that each variant was inherited from one parent, consistent with the compound heterozygous state in the proband. We hope this addition meets the reviewer's expectations regarding data validation and transparency.
Page 10 “Primers were designed to amplify a fragment of the SERPINF1 gene flanking the c.261_265dup variant. Two TaqMan probes were designed to anneal to the wild-type and mutant alleles, respectively”. Please add primer sequences to analyze them.
Response:
We thank the reviewer for this valuable suggestion. In response, we have added the primer sequences used for the allele-specific real-time PCR analysis of the c.261_265dup variant in the SERPINF1 gene to the Materials and Methods section of the manuscript. The reactions were performed in duplicate using two independently designed primer pairs to ensure specificity and reproducibility. The following sequences have been included:
· SERPINF1_261_265dup_RT_1_F: AACGTGCTCCTGTCTCCTCT
· SERPINF1_261_265dup_RT_1_R: GAGCACTCACCCAGCGAG
· SERPINF1_261_265dup_RT_2_F: CTGTCTCCTCTCAGTGTGGC
· SERPINF1_261_265dup_RT_2_R: GCTTCCTGCATCTGAGCACT
We hope this addresses the reviewer’s concern.
Page 6 “The diagnosis of OI type VI was confirmed in the examined patients by sequencing a targeted gene panel including genes responsible for OI”. Please describe in the Materials and Methods section the gene panel built for NGS analysis.
Response:
Thank you for your valuable comment. We have added a detailed description of the gene panel used for NGS analysis in the Materials and Methods section. Specifically, we clarified that the diagnosis was confirmed by sequencing the targeted panel entitled “Hereditary Diseases with Skeletal Pathology”, which includes 166 genes associated with hereditary skeletal disorders. The full list of genes included in this panel is now provided in the Supplementary List 1.

Reviewer 2 Report
Comments and Suggestions for Authors
Please find my comments attached in the PDF file

Author Response
Response to the Reviewers We sincerely thank the reviewers for their detailed and insightful comments, which helped us to improve the clarity and quality of our manuscript. We have addressed all the suggestions and concerns raised, and corresponding changes have been made in the revised version of the manuscript. Below, we provide a detailed point-by-point response to each reviewer’s comments.
Major comments:
- While the authors report novel SERPINF1 gene variants associated with osteogenesis imperfecta type VI including c.185G>T (p.Gly62Val) and c.992_993insCA (p.Glu331Asnfs) a critical limitation is the absence of visual confirmation and documentation of the sequencing data. For example: NGS or Sanger sequencing screenshots are provided to support the identification of the novel variants. Inclusion of such Figures is standard and essential for validation and transparency, particularly when reporting novel variants. Additionally, the exact nucleotide sequence and genomic context of the novel variants are not provided in the main text or as supplementary material.
Please include in the manuscript or supplementary material, Figures for: annotated NGS or Sanger sequencing screenshots for both novel variants (e.g., IGV plots or chromatograms).
The full cDNA or protein sequence highlighting the mutation site, preferably with reference alignment to wild type.
If feasible, provide a Lollipop plot or domain map of the PEDF protein showing the position of both mutations relative to known functional domains.
Response:
We thank the reviewer for this helpful comment. In response, we have added a new figure to the manuscript, presenting Sanger sequencing chromatograms that confirm the presence of two variants in the SERPINF1 gene:
- 185G>T (p.Gly62Val) — in the proband and his mother (panels A and B),
- 992_993insCA (p.Glu331Asnfs) — in the proband and his father (panels A and C).
Red arrows indicate the variant positions, while blue arrows highlight the parental inheritance. These data confirm that each variant was inherited from one parent, consistent with the compound heterozygous state in the proband. We hope this addition meets the reviewer's expectations regarding data validation and transparency.
In the main body of the manuscript, we have provided a detailed description of the identified variants, including their nomenclature, genomic and protein-level localization, predicted molecular consequences, and classification according to ACMG guidelines. Specifically, we discussed how the c.992_993insCA (p.Glu331AsnfsTer3) variant leads to a truncated PEDF protein without triggering nonsense-mediated decay, and how the c.185G>T (p.Gly62Val) variant may affect the protein’s structural stability and secretory function. These interpretations are supported by transcriptomic analysis and are presented in detail in Section 2.2 and the Discussion.
Given that the structural and functional consequences of both variants are extensively described in the text, we believe that an additional lollipop plot or protein sequence alignment would not substantially enhance the understanding of our findings. However, we are happy to provide such supplementary materials upon editorial request.
- Authors mention the use of OMIM, LOVD, gnomAD, etc., these should be cited formally with proper references in the bibliography. Hyperlinks in the text are insufficient. For example, information in the introduction section about OMIM database (https://www.omim.org/). Have to include as reference and the relevant information in the web plot as summary table. This can be placed in the Supplementary Material or manuscript. Additionally, The hyperlink to the Thermo Fisher TaqMan protocol (https://www.thermofisher.com/content/dam/LifeTech/Documents/PDFs/Manuals/tqma n-allelic-discrimination-assay-manual.pdf. Accessed March 7,2023) provide in the manuscript doesn’t exist. It should either be corrected or removed.
Response:
We thank the reviewer for this important suggestion.
- All databases mentioned in the manuscript (OMIM, LOVD, gnomAD, 1000 Genomes Project, ESP6500, HGMD Professional) are now cited with formal bibliographic references, which have been added to the reference list in accordance with MDPI formatting guidelines.
- We have created a Supplementary Table (Table S1) summarizing each database, including its full name, URL, and a brief description of its contents and relevance. This table has been included in the revised Supplementary Materials.
- The previously included non-functional hyperlink to the TaqMan® Allelic Discrimination Assay User Guide has been removed and replaced with a functional and properly cited reference to the official Thermo Fisher Scientific manual:
Applied Biosystems. TaqMan® Allelic Discrimination Assay: User Guide; Thermo Fisher Scientific: Waltham, MA, USA, 2013. Available online: https://tools.thermofisher.com/content/sfs/manuals/cms_041280.pdf (accessed on May 22, 2025).
- Consider a phenotype-genotype correlation table or graphic summarizing all identified variants and associated features.
Response:
We thank the reviewer for this valuable suggestion regarding the inclusion of a phenotype-genotype correlation table. We fully agree that such a summary enhances the clarity and impact of our findings.
In the revised manuscript (see Table 1), we have included a comprehensive table presenting the clinical and radiological characteristics of all four patients, including genotype, age at first fracture, total number of fractures, height (SD score), ambulatory status, bone mineral density, and the severity of long tubular bone deformities. This provides a clearer overview of the relationship between the identified SERPINF1 variants and the observed phenotypes.
While we acknowledge the small cohort size, we believe that, given the rarity of osteogenesis imperfecta type VI and the presence of both novel and previously reported variants, the presented data illustrate meaningful trends that may suggest genotype–phenotype correlations. We have added a statement to the manuscript emphasizing the preliminary nature of these observations and the need for validation in larger cohorts.
- The Materials and Methods section is overloaded. It would benefit from being clearly divided into subsections (e.g., "Clinical Evaluation", "Radiographic Analysis", "Genetic Sequencing", "RNA Analysis", "Population Study").
Response:
We thank the reviewer for this valuable suggestion. In response, we have substantially revised the Materials and Methods section to improve clarity and readability. It is now clearly structured into the subsections:
These structural improvements were made in accordance with the journal’s formatting guidelines and are intended to enhance the manuscript’s overall accessibility to readers. We believe that this reorganization provides a clearer understanding of the methodological workflow and better supports the validity of our findings.
- The study would benefit from an original figure illustrating the methodology, especially the sequencing and variant analysis pipeline.
Response:
We thank the reviewer for the valuable suggestion regarding the inclusion of a figure illustrating the methodology. In response, we have added a new figure (Figure 4) that schematically represents the study workflow, including the steps of DNA extraction, library preparation, targeted panel sequencing, variant calling, validation, and interpretation. This diagram enhances the transparency and clarity of the methods used in our genetic analysis of SERPINF1 variants. The figure has been placed in the Materials and Methods section (Section 4.3 “Genetic Sequencing”).
- Reverse transcription was conducted with the Reverse Transcription System (Dialat, Russia), following the manufacturer’s protocol. Have to be describe.
Response:
We thank the reviewer for this valuable comment. In response, we have significantly expanded the description of the reverse transcription procedure in the Methods section (Subsection 4.5 “RNA Analysis of the Variant c.992_993insCA”). The revised text now includes detailed information on the composition of the reaction mixture, incubation conditions, and storage of the synthesized cDNA. These additions ensure reproducibility and provide clarity for future studies. We believe this revision enhances the transparency and methodological rigor of our work.
Minor comments:
- The manuscript lacks line numbers, which are essential for reviewing and referencing specific parts of the text.
Response:
Thank you for your comment. We would like to clarify that line numbering is present in the submitted manuscript and was formatted according to the journal’s submission guidelines. It is possible that a technical issue during PDF rendering or the viewing software may have affected the display of line numbers. We will double-check the file prior to resubmission to ensure that line numbers are clearly visible to all reviewers.
- In the section reporting the estimated allele frequency of the c.261_265dup variant in the Tuvan population, the authors state: “The mutant allele frequency was calculated using the Hardy–Weinberg equation. The 95% confidence intervals (95% CI) for the mutant allele and OI frequencies were calculated using the Clopper–Pearson exact method [6].” https://www.jstor.org/stable/2331986?seq=1. However, the equation itself is not shown, and the methodology used to perform the calculations is not adequately described. Furthermore, it is unclear how the Clopper–Pearson method was specifically applied (e.g., input parameters, assumptions, software/tool used).
Response:
The prevalence of OI type VI was calculated using the Hardy–Weinberg equilibrium formula (1):
p² + 2pq + q² = 1 (1)
where 2pq is the expected frequency of heterozygotes,
and q is the allele frequency of the variant under investigation.
The observed heterozygous carrier frequency of the pathogenic SERPINF1 variant was calculated using equation (2):
2pq = m / 2N (2)
where 2pq is the proportion of heterozygotes,
m is the number of identified heterozygotes,
and N is the total number of individuals screened.
Thank you for pointing out the need to clarify the application of the Clopper–Pearson method. In the revised manuscript, we now indicate that the 95% confidence intervals were calculated using the BinomCI function from the R package DescTools (Signorell A, 2025).
This method was chosen because the presence or absence of a variant (or disease) can be modeled as a binomial process. In the case of the carrier frequency, we treated 7 heterozygous individuals among 801 healthy newborns as 7 "successes" in 1602 trials (two chromosomes per individual).
For the disease frequency, the total population size of the Republic of Tuva in 2024 (337,554) was multiplied by q², the estimated frequency of affected individuals, to derive the input for binomial CI calculation.
- Table 1 must be revised to conform to MDPI style requirements.
Response:
Thank you for your valuable comment. Table 1 has been revised in accordance with the MDPI formatting guidelines. All column headers are now bold and centered, units are included in the headings, abbreviations are explained in the footnote, and the layout follows MDPI’s minimalist table style.
- The title could be more specific by mentioning the novel variants and the ethnic context
Response:
We appreciate the reviewer’s suggestion. However, we believe that the current title — “Molecular and Clinical Aspects of Osteogenesis Imperfecta Type VI” — accurately reflects the scope of the study, which includes both clinical observations and molecular-genetic analysis of this rare OI subtype. Including specific gene variants or the ethnic background in the title would unnecessarily narrow the focus, while these important details are thoroughly addressed in the abstract and the main text.
- Graphical abstract would be helpful to summarize the study visually
Response:
We thank the reviewer for this valuable suggestion. In response, we have prepared and included a graphical abstract that summarizes the key aspects of our study, including the clinical features, identification of two novel SERPINF1 variants (c.185G>T and c.992_993insCA), the founder effect of the c.261_265dup variant in the Tuvan population, and the main conclusions regarding genetic screening. The visual elements were carefully selected to enhance clarity and accessibility for a broad scientific audience. We believe this graphical abstract effectively supports the manuscript content and improves its presentation.
- Define “SD” and all other abbreviations that lacks mentions.
Response:
Thank you for the comment. We have added definitions at first mention for all previously undefined abbreviations, including:
- SD — standard deviation
- Z-score — standard deviation score used in densitometry
- CI — confidence interval
These have now been clarified in the revised manuscript to ensure clarity for all readers.
Round 2
Reviewer 2 Report
Comments and Suggestions for Authors
The manuscript has been significantly improved. I accept the responses and consider the manuscript ready for acceptance, with only minor considerations regarding the supplementary materials:
The authors have to add a cover page including the title, authors, and affiliations.
The supplementary materials currently use three different font styles (Times New Roman, Calibri, and Cambria) that need to be standardized. Additionally, the table formatting should be updated to comply with MDPI format requirements.
Finally, in the manuscript, Equation is used more than formula; consider changing that.
Author Response
Dear Reviewer,
We would like to sincerely thank you for your thorough evaluation of our manuscript and for your constructive suggestions, which have significantly improved the quality of our supplementary materials.
Please find below our responses to your comments:
- Title page:
As requested, we have added a title page to the Supplementary Material. It now includes the full title of the manuscript, the list of authors, their affiliations, and the corresponding author's contact information.
- Font unification and table formatting:
All fonts in the Supplementary Material have been standardized to Times New Roman, 12 pt, including headings, table contents, and footnotes. In addition, all tables have been reformatted to align with MDPI formatting standards—ensuring consistency in style and appearance.
- Terminology (“formula” to “equation”):
We have replaced the word “formula” with “equation” in the context of Hardy–Weinberg calculations to ensure consistency with terminology used in the main manuscript and MDPI guidelines.
We hope that these changes fully address your concerns. We are grateful for your valuable feedback and the opportunity to improve our work.
With kind regards,
Elena Merkuryeva
(on behalf of all co-authors)
Corresponding author
elena.merkureva@gmail.com
